# Intraguild Predation between *Chrysoperla carnea* (Neuroptera: Chrysopidae) and *Hippodamia variegata* (Coleoptera: Coccinellidae) at Various Extraguild Prey Densities and Arena Complexities

**DOI:** 10.3390/insects11050288

**Published:** 2020-05-08

**Authors:** Maryam Zarei, Hossein Madadi, Abbas Ali Zamani, Oldřich Nedvěd

**Affiliations:** 1Department of Plant Protection, Faculty of Agriculture, Bu-Ali Sina University, Hamedan 6517838695, Iran; zarei.ma68@gmail.com; 2Department of Plant Protection, College of Agriculture, Razi University, Kermanshah 6714414971, Iran; azamani@razi.ac.ir; 3Faculty of Science, University of South Bohemia, Branišovská 1760, 37005 České Budějovice, Czech Republic; nedved@prf.jcu.cz; 4Biology Centre, Czech Academy of Sciences, Institute of Entomology , Branišovská 31, 37005 České Budějovice, Czech Republic

**Keywords:** predator-predator interaction, black bean aphid, green lacewing, lady beetle, competition

## Abstract

Intraguild predation (IGP) is a ubiquitous, important and common interaction that occurs in aphidophagous guilds. The effects of extraguild prey (EGP, i.e., aphids) density, predator life stage combinations and duration of the interaction on the level, asymmetry and direction of intraguild predation between lacewing *Chrysoperla carnea* and ladybird *Hippodamia variegata* were examined in simple laboratory arena and more complex microcosm environment. Three initial densities of 50, 150 and 400 *Aphis fabae* third instar nymphs and a control without aphids were provided to six combinations of predator life stages (2nd and 3rd larval instars of lacewing and 3rd and 4th instars and adult females of ladybird). The remaining aphid density and occurrence of IGP were checked after 24, 48 and 72 h. The IGP intensity (IGP level, IL) was similar in the simple arena (reaching 0.6 between larvae in absence of EGP and 0.3 between lacewing larvae and ladybird females) and microcosm environment (0.3 without EGP). In both environments, increasing EGP density lowered IL according to negative exponential relationship. IGP was asymmetric (general average asymmetry was 0.82 in simple arena and 0.93 in microcosm, the difference was not significant) and mostly in favour of larvae of *C. carnea*, except in the combination of 2nd larvae of *C. carnea* with the 4th larvae and adults of *H. variegata.* The direction of IGP, but not other characteristics, partially changed during the duration of the experiment. The incidence of IGP interactions among aphid predators under real conditions and its consequences on aphid biological control are discussed.

## 1. Introduction

Two major interactions between animals occurring together in nature are competition and predation, which sometimes combine as intraguild predation (IGP) when potential competitors for a shared prey (the extraguild prey: EGP) switch to the predation of each other [1]. In every IGP interaction, there is a top predator (aggressor) which is called intraguild predator (IG predator) and one or more predatory species (intraguild prey or IG prey) which have been attacked and killed by top predator. IGP, which is a widespread interaction in diverse guilds encompassing polyphagous predators and to a lesser extent, specialists, may affect spatial distribution, searching behaviour and abundance of the involved species [2]. Theoretically, it has been hypothesized that IGP occurrence can lead to different effects ranging from exclusion to coexistence of engaged species [1,3]. Some models proposed that the coexistence of both predators occurs if the IG predator is inferior at exploitation of shared resource than IG prey. Conversely, if the IG prey was inferior at exploitation, the top predator drives it to exclusion [3,4]. IGP increases the fitness of superior IG predator [1] although it remains unclear if this is a result of the additional food source or of the elimination of a potential competitor. The diet rule says that the top predator feeds on the less profitable of its two prey species (mostly the occasional IGP rather than the primary EGP) only if the more profitable one (EGP) is rare [5]. 

IGP has been demonstrated amongst beneficial insects used for biological control [2,6,7,8], including aphidophagous natural enemies (e.g., [9,10,11,12,13,14]). IGP rarely disrupt biological control in programs targeting the same pest species with multiple natural enemies [15].

Several natural enemies of aphids form a diverse aphidophagous guild. Because most of them are polyphagous, the likelihood of IGP incidences is high. Two major predators of the black bean aphid *Aphis fabae* that can meet each other in the aphid colonies are green lacewing *Chrysoperla carnea* (Stephens) (Neuroptera: Chrysopidae) [16] and the variegated ladybird *Hippodamia variegata* (Goeze) (Coleoptera: Coccinellidae). These two species are among the most common aphid predators in many agroecosystems ([17,18] and several observations by the senior author), although their contemporary occurrence is seldom reported due to taxonomic restrictions of field surveys. The green lacewing is predacious in the larval stage whereas both larvae and adults of the ladybird are predators. Both species are thus prone to be involved in IGP interactions, both with each other and with other species in the aphidophagous guild [19,20,21]. The IGP incidence and outcome between different ladybird species (e.g., *Harmonia axyridis* (Pallas), *Coleomegilla maculata* (DeGeer) and *Adalia bipunctata* (L.)) and green lacewings is documented [20,22] but until now no study considered the IGP interaction between *C. carnea* and *H. variegata*.

Several variables including the life stages of IG predators and preys, complexity of experimental arenas and EGP quantity and quality influence IGP intensity, symmetry and direction [9,10,23,24,25,26]. IGP is sometimes strongly or completely asymmetric, as a rule if one of the species is a parasitoid or predominantly when an insect with chewing mouthparts and a hard cuticle (such as ladybirds *H. axyridis* or *A. bipunctata*) kills an insect with stylets or proboscis and a soft cuticle (such as hoverfly *Episyrphus balteatus* (DeGeer) or true bug *Macrolophus pygmaeus* (Rambur) [8,9]).

The effect of EGP density on IG predator seems to be mediated by satiation of the IG predator, so that predation on the IG prey decreases [27] which prevents its exclusion from that patch. Several studies with aphidophagous predators have addressed IGP and its outcome and intensity under various conditions but most have been carried out under small-scale laboratory conditions, making extrapolation to field conditions difficult. Those studies with various levels of habitat complexity have shown that IGP intensity diminishes with increasing size or complexity of experimental arena [23,28]. Other studies demonstrated that EGP density diminished IGP intensity [25]. However, few studies considered both variables alongside different life stages of species engaging in IGP. 

The aim of this research was to explore the effect of EGP (aphid) density, experimental arena complexity (simple arenas and complex microcosms with similar volume), duration of their interaction and developmental stages of two aphid predators *C. carnea* and *H. variegata* on their IGP intensity, symmetry and direction. We involved several mobile developmental stages with high predation rate but contrasting body size and mechanical protection (compare to [26]). We predict a decrease of IGP level with increasing EGP density and with increasing arena complexity.

## 2. Materials and Methods

### 2.1. Prey and Predators Rearing

*Aphis fabae* was collected from broad bean fields in Hamedan Province, Iran and established on potted broad bean (*Vicia faba* L., c.v. Barekat) grown in a greenhouse at 25 ± 5 °C, 50 ± 10% RH and 16:8 L:D. Adults of *C. carnea* and *H. variegata* were collected from alfalfa fields in Hamedan Province, Iran, and kept separately. All rearings and experiments have been described in our previous study [29]; they took place in a climate controlled chamber at 25 ± 1 °C, 60 ± 10 % RH and 16:8 L:D. The lacewings were placed in oviposition containers (6.5 × 11.5 cm = diameter × height) and supplied with a mixture of yeast, water and honey. The ladybirds were kept in ventilated Petri dishes (9 cm diameter), fed daily ad libitum with *A. fabae* and eggs of Mediterranean flour moth, *Ephestia kuehniella* Zeller, and provided with honey solution (10%) as supplemental food.

Predators laid eggs on the provided oviposition substrate (pieces of corrugated paper for ladybeetles and black cardboard for lacewings). Eggs were collected daily (lacewings) or every 8 h (ladybirds) and transferred to separate Petri dishes (diameter: 3 cm; height: 1 cm). Dishes were checked for egg hatching every 12 h and larvae were transferred to individual Petri dishes. Larvae were fed with *A. fabae* ad libitum until they reached the required stage. Both predators were reared for five generations prior to the experiments.

### 2.2. Experimental Design

All six combinations of one larva of the second (L2) or third instar (L3) of *C. carnea* (Cc) and one larva of the third (L3) or fourth (L4) instar or adult female (A) of *H. variegata* (Hv) were used, ten replications per combination. The mean body length of the 2nd and 3rd lacewing larvae were 3.76 ± 0.09 and 6.09 ± 0.12 mm while the third and fourth instar ladybird larvae were 4.42 ± 0.15 and 5.86 ± 0.12 mm long. The ladybird females were three days old and mated. All predators were starved for six hours prior to the experimentation to increase their motivation to search for prey. The first instar of Cc and the first and second instars of Hv were excluded from the experiments due to their relatively low predation rate. 

To assess the influence of EGP density on IGP intensity (level), symmetry and direction, three initial aphid densities selected after previous assays [29] were used: 50, 150 and 400 third instar nymphs of *A. fabae* together with a series without aphids (0). The surviving prey was counted and was not replenished during the experimental period. Thus, the initial density of aphids on subsequent days was the number of alive aphids remaining in each individual arena. Killed prey was punctured and sucked-up or completely consumed and thereby distinguished from natural mortality. 

To assess the influence of arena complexity, two experimental set-ups were used. (1) Simple arena consisted of a ventilated plexiglass container (about 12 (l) × 9 (w) × 4 (h) cm) with black bean aphids feeding on two broad bean leaves placed upside down on a moistened cotton wool and petioles covered with cellophane. The two predators were placed on their own leaves. (2) Microcosm arena: two transparent plastic containers (drink cups), each 12 cm height × 7 cm diameter; bottom one with soil and a potted broad bean seedling (about 10 cm high with 4 to 6 leaves, see also [8]) with black bean aphids evenly distributed on leaves; top one ventilated, enclosing the seedling. The predators were placed randomly on the plants.

All experimental units were checked nondestructively after 24, 48 and 72 h. The state of the two predators was recorded to determine IGP intensity (level), symmetry and direction. Ten replications per combination of predator stage and aphid density were used at both set-ups. 

### 2.3. Level, Symmetry and Direction of IGP

Level or intensity of IGP (IL) was defined as the number of predators killed (Nk) divided by the initial number (Ni) of pairs of predators that interact during experimental day: IL = Nk/Ni. It is the number of replicates which IGP occurred divided by total number of replicates [10,26], thus ranging from 0 to 1. Asymmetry of IGP (AI) represents the number of killed individuals of one predator species (Na) divided by the number of all killed predators (Nk) that day, adjusted in a way to cover values from 0 (full symmetry) to 1 (extreme asymmetry): AI = 2*abs(Na/Nk − 0.5). It is thus independent on the initial number of predators that day. If no predator was killed within the treatment and day, the value was set as missing.

To determine the IGP direction between different stages of the two predators, direction index (DI) of IGP was used, which is defined as the ratio of the number of replicates in which a given predator was killed (Na) with the total number of replicates (Nk) in which IGP occurred; DI = Na/Nk [10]. It ranges from 0 (predator a always wins) to 1 (predator b always wins), with value 0.5 meaning IGP symmetry (AI = 0). 

### 2.4. Statistical Analysis

We used general custom designs (GLZ) analyses with Tweedie distribution and log link function for variables IL, AI and DI that are not normally distributed. GLZ evaluated two categorical (predator stages) and two continuous variables (EGP density and time = day of observation). The same model was used separately for each experimental arena and also to compare between the two experimental set-ups = arena complexities. Nonlinear regression using negative exponential equations was computed to model the decrease of IL with EGP density: IL=a*exp(b*Aphids), b < 0, estimated by Levenberg–Marquardt method. All tests were performed in Statistica 13 software [30].

## 3. Results

### 3.1. IGP Level and Symmetry

In the simple arena, time (day of experiment) and stage of lacewing had no effect on IL (*χ*^2^ = 0.0026, df = 1, *p* = 0.96 and *χ*^2^ = 2.1, df = 1, *p* = 0.15) respectively, while the effect of number of aphids (EGP) was significant very strong (*χ*^2^ = 52, df = 1, *p* < 10^−6^). Stage of ladybird influenced IL significantly (*χ*^2^ = 9.9, df = 2, *p* = 0.0072) and it was higher with the two larval instars and lower with adult ladybird. Thus, we plotted the relationship of IL and aphid density separately for *H. variegata* larvae and adults (Figure 1). The parameters of the negative exponential relationship IL = a*exp(b*Aphids) for larvae 3 and 4 were a = 0.60 ± 0.05, b = −0.054 ± 0.016, R^2^ = 0.63; for adults they were a = 0.33 ± 0.04, b = −0.119 ± 0.069, R^2^ = 0.63.

Additionally, time (day of experiment)(*χ*^2^ = 0.018, df = 1, *p* = 0.89); aphid density (*χ*^2^ = 0.62, df = 1, *p* = 0.43); stage of lacewing (*χ*^2^ = 2.3, df = 1, *p* = 0.13) and stage of ladybird (*χ*^2^ = 1.4, df = 2, *p* = 0.49) did not have significant effect on AI. General average asymmetry was 0.82. Average number of pairs of predators per treatment remaining in the experiment after one day was 8.7 (range 2–10) and after two days 7.1 (0–10).

In microcosm set up, time (day of experiment) had no significant effect on IL (*χ*^2^ = 1.0, df = 1, *p* = 0.32). The effect of number of aphids was very strong (*χ*^2^ = 38, df = 1, *p* < 10^−6^). Neither the stage of lacewing (*χ*^2^ = 0.0048, df = 1, *p* = 0.94) nor the stage of ladybird (*χ*^2^ = 0.15, df = 2, *p* = 0.93) had any effect on IL. The relationship of IL and aphid density (Figure 2) had negative exponential shape IL = a*exp(b*Aphids) with parameters a = 0.30 ± 0.03, b = −0.020 ± 0.006, R^2^ = 0.41.

None of the variables (day of experiment, stages of lacewing and ladybird and initial number of aphids) had significant effect on AI (*χ*^2^ < 0.99, *p* < 0.32). General average asymmetry was 0.93. Average number of pairs of predators per treatment remaining in the experiment after one day was 9.2 (range 5–10), after two days 8.3 (4–10).

Then we compared the data from simple arena and microcosm with categorical factor stage of ladybird and continuous factor aphid density (since stage of lacewing and time of experiment were not significant in both separate analyses). Only the density of aphids had significant effect on IL (*χ*^2^ = 79, df = 1, *p* < 10^−6^). The effect of arena was not significant (*χ*^2^ = 0.18, df = 1, *p* = 0.67) and neither was the effect of ladybird developmental stage (*χ*^2^ = 4.3, df = 2, *p* = 0.12). None of the factors (arena, ladybird stage, aphid density) had any effect on AI (*χ*^2^ < 0.62, *p* < 0.42).

### 3.2. IGP Direction

In contrast to IL, the IGP direction (DI) was independent on the aphid density (*χ*^2^ = 0.40, df = 1, *p* = 0.52) and dependent on the time of experiment (*χ*^2^ = 8.9, df = 1, *p* = 0.0028), lacewing stage (*χ*^2^ = 12, df = 1, *p* = 0.0060) and ladybird stage (*χ*^2^ = 14, df = 2, *p* = 0.00075) in the simple arena.

Second instar lacewing larvae were not able to kill adult ladybirds and were moderately killed by them equally during the three experimental days (Figure 3). They killed a moderate number of third instar ladybird larvae. Surprisingly, they killed a high number of fourth instar ladybird larvae during the first day but were killed by the surviving ladybird larvae during the second and third days. 

There was a low and symmetric IGP between the third instar lacewing larvae and adult ladybirds (Figure 4). However, old lacewing larvae killed many ladybird larvae of both third and fourth instars. Ladybird larvae killed small numbers of lacewing larvae.

DI in microcosm was independent on the aphid density (*χ*^2^ = 0.73, df = 1, *p* = 0.39) and on the time of experiment (*χ*^2^ = 0.035, df = 1, *p* = 0.85). It was dependent on lacewing stage (*χ*^2^ = 9.1, df = 1, *p* = 0.0025) and ladybird stage (*χ*^2^ = 18, df = 2, *p* = 0.00010; Figure 5).

Second instar lacewing larvae were not able to kill adult ladybirds and old ladybird larvae and were moderately killed by them (Figure 5). They killed a moderate number of third instar ladybird larvae. Third instar lacewing larvae were not able to kill adult ladybirds and were moderately killed by them, while they killed a high number of third and fourth instar ladybird larvae.

## 4. Discussion

### 4.1. Prey Density 

Our study confirmed previous findings that predator life stages and EGP density have an important effect on the IGP intensity (IL), while the role of the time that the predators have been exposed to each other played a limited role [8,20]. Contrastingly, the IGP symmetry (AI) and direction (DI) were independent on the prey density, while they strongly differed between the combinations of developmental stages of the two predators. In the simple arena, DI also changed over time. 

High aphid density substantially decreased the number of predators killed. The level of IGP was generally high in the absence of EGP, especially in the simple arena and interaction of ladybird larva with lacewing larva. The presence of EGP substantially decreased the level of IGP in such an extent that at the initial density of 400 black bean aphids, no intraguild predation occurred. This high-EGP treatment also substitutes experimental control, as there was no natural mortality observed. Predator pairs that included adult ladybirds showed lower IL than combinations with larvae. IL was highest in the combination of the third larval instars of the predators in the simple arena. The lacewing larvae were the intraguild predator and the third instar ladybirds tended to be the intraguild prey in this combination.

The increase of EGP density in several studies had a diluting effect on intensity of IGP even if the superior intraguild predators experienced little risk when confronted with the inferior predator, i.e., intraguild prey [10]. Theoretically, it has been modelled that an optimal foraging predator might avoid the IG prey as its potential diet when density of extraguild prey increased [4]. High EGP densities of *Tetranychus urticae* Koch (Acari: Tetranychidae) and *Aphis gossypi* Glover (Hemiptera: Aphididae) decreased the IGP incidence but did not change the direction of IGP between *H. axyridis* and *Coccinella undecimpunctata* L. [24,31]. IGP between *Coccinella septempunctata* L. and *H. variegata* larvae was reduced under high density of *A. gossypi* [32]. However, unlike our results, the EGP density did not affect lacewing survival when confronted with the invasive ladybird *Harmonia axyridis* [14]. The possible reasons for severe reduction of IGP intensity at high abundance of EGP is the change of behaviour of both the IG prey and IG predator. When EGP density is low, the searching for it increases in both predators and, as a consequence, the confrontation frequency between them increases.

Changes of IL and DI over time in the experiment can also be mediated via the EGP density. When the main EGP (pest species) becomes scarce in nature, the predators broaden their food ranges to encompass other species including IG preys [10]. Thus, we expected that IL should increase with experiment progression when the initial density of aphids was high but gradually decreasing because the aphids eaten were not replenished. In the absence of EGP, the IL should be constant over time. In the analysis, the day of observation showed no effect, because all changes (increased IL) were explained simply by the EGP density (its decrease over time). Treatments that included the fourth ladybird instar did not increase IL but strongly changed DI—lacewing larvae won during the first day, when aphids were dense, but failed during the last day, when EGP became scarce. This is a surprising result when compared to the study of *C. carnea* larvae wining over *H. axyridis* larvae in the absence of aphids [12]. 

### 4.2. Habitat Complexity

In other studies, it has been repeatedly demonstrated that in more complex and three-dimensional searching habitats, the encounter rate of an intraguild predator with intraguild prey reduced [33], sometimes only for specific developmental stage [8]. Increasing the complexity of our experimental arena (using broad bean seedlings as experimental environment instead of separate leaves, without substantial increase of the arena volume) only slightly decreased IL. Habitat complexity did not substantially affect AI and DI between *C. carnea* and *H. variegata*. Although there was lower average IL in microcosm than in the simple arena (compare Figure 1 and Figure 2), this difference was mediated through EGP density. Because aphids were more easily found and eaten in the simple arena, the aphid density on the second and third days was much lower than in microcosm [21]. Similarly, the invasive crayfish *Procambarus clarkii* achieved much higher maximum prey eaten on plane surface than on sandy bottom and the least on gravel bottom [34]. Low aphid density in our simple arena corresponded to higher IL. Thus, direct effect of the arena complexity on IGP was low. This means that there was no difference between arenas in the difficulty to meet the other predator. 

However, we observed that ladybird larvae tended to avoid confrontation with lacewing larvae. Thus, in the absence of aphids, they were scattered around the pot or they were concealed in some parts of faba bean seedlings. Such avoidance behaviour has been noted in several studies [35]. Although *C. carnea* and larvae of *C. septempunctata* killed each other in Petri dishes, they did not interact on potted faba bean seedlings [36]. 

### 4.3. Body Size and Aggressiveness

One of the factors known to be responsible for IGP symmetry and direction is body size. Accordingly, larvae of *C. septempunctata* killed larvae of *H. variegata* at a higher rate (0.70) than vice versa (0.43) [32]. In our experiments, in the combination of the 2nd instar larvae of *C. carnea* with the 4th instar larvae of *H. variegata*, the smaller green lacewing larvae were the intraguild prey. The opposite situation appeared between the 3rd instars: the 3rd instar lacewing larvae were almost 1.5 times as great as that of the 3rd ladybird larvae, and there were also mobility differences. For an equivalent predator size, the lacewings won the contest with ladybird larvae in several studies [10,19,21]. In our previous study, we have shown prevalent IGP dominance of L3 larvae of *C. carnea* over much bigger L4 larvae of ladybird *H. axyridis* [12]. 

The relatively large body is associated with greater power and more effective defence mechanism [34]. Moreover, older 4th instar ladybird larvae stop feeding and then become motionless which can change the direction of IGP regardless of body size. Interaction of soft larvae of any species with hard sclerotized adult ladybirds can result in predation of larva by the adult regardless of the body size difference. The mobility of predators was also considered as a critical factor of IGP direction [2,10]. Sometimes the smaller larva could act as an IG predator because of their more aggressive habits: the less mobile grown ladybird larvae (several days after moulting from the previous instar) have been more frequently attacked by green lacewing larvae. The greater success of *H. axyridis* than *C. septempunctata* has been attributed to the high rate of attack and escaping ability [37]. In the absence of prey, *C. carnea* third instars attacked and killed fourth instar larvae of *C. maculata* [10,20]. This aggressive habit was also reported for other lacewings [21].

We observed that after releasing predators in the experimental arena, they started to search for food. When *C. carnea* larva encountered the 3rd instar larva of *H. variegata* in the absence of EGP, it stabbed the sickle mouthparts to the soft abdominal segments of *H. variegata* larva. The ladybird larva tried to defend itself with rapid jerking of abdomen but this behaviour was not effective. However, in the previous study [12] larvae of ladybird *H. axyridis* looked paralyzed immediately after being bitten by *C. carnea* larva. Moreover, the long parascoli with hairs and alkaloids of ladybird larvae seemed to be noneffective against lacewing larvae [12]. It seems that the lacewings should be able to detoxify also the alkaloid hippodamine found in *H. variegata* larvae. 

The sickle mouthparts of lacewing larvae enable them to be more aggressive and win in IGP. Larvae of *C. carnea* attacked *H. variegata* larvae more often in the posterior abdominal segments, especially their ventral side, which is thinner and more vulnerable than tergites. This position facilitates the feeding of lacewing larvae from ladybird victims with a lower risk of being counter-attacked. 

## 5. Conclusions

Intraguild interactions among polyphagous natural enemies are composed of two mechanisms, competition and predation. Intraguild interaction between *C. carnea* and *H. variegata* seems to be mostly of the competitive type. We demonstrated that IGP intensity decreased with increasing extraguild prey density rather than with increasing arena complexity. Thus, the question whether IGP threat under real field conditions would be lower than predicted by voluminous laboratory studies in simple arenas remains unresolved. We predict that lacewing and ladybird larvae could survive together without deleterious effects on biological control of aphids if the prey is abundant. 

## Figures and Tables

**Figure 1 insects-11-00288-f001:**
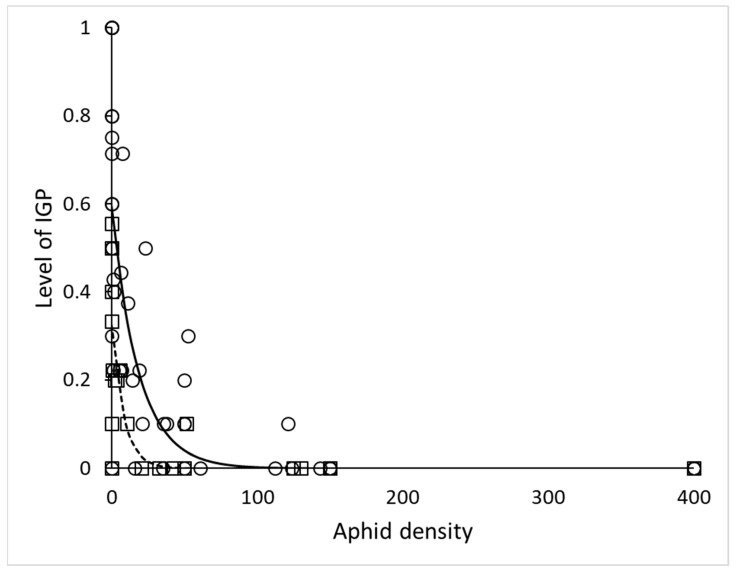
Decrease of the intraguild predation level with increasing density of extraguild prey (EGP) in the simple arena. Circles and solid line: treatments containing larvae of *Hippodamia variegata* and *Chrysopa carnea*, y = 0.60*exp(−0.054*x), N = 48; squares and dashed line: treatments containing adults of *H. variegata* and larvae of *C. carnea*, y = 0.33*exp(−0.119*x), N = 24.

**Figure 2 insects-11-00288-f002:**
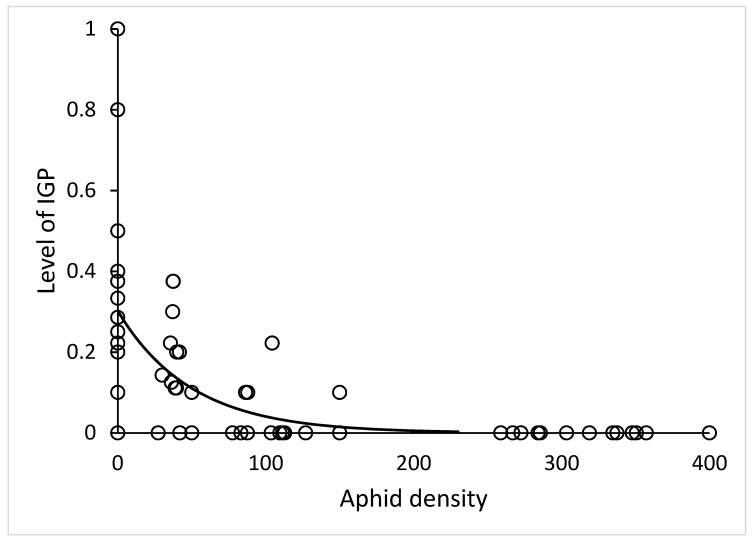
Decrease of the intraguild predation level with increasing density of EGP in a microcosm: y = 0.30*exp(−0.020*x), N = 72.

**Figure 3 insects-11-00288-f003:**
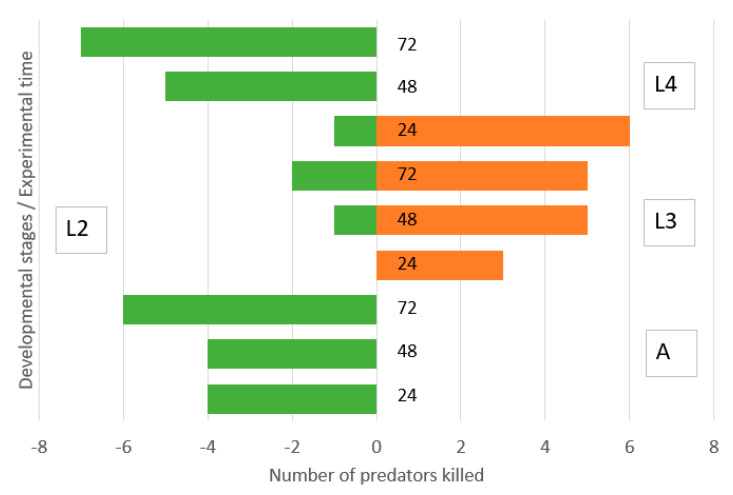
Direction of intraguild predation between *Chrysoperla carnea,* second instar larvae, and *Hippodamia variegata* adults (A), third instar larvae (L3) and fourth instar larvae (L4) in the simple arena. Numbers of predators killed of initial N = 40 during 0–24, 24–48 and 48–72 h.

**Figure 4 insects-11-00288-f004:**
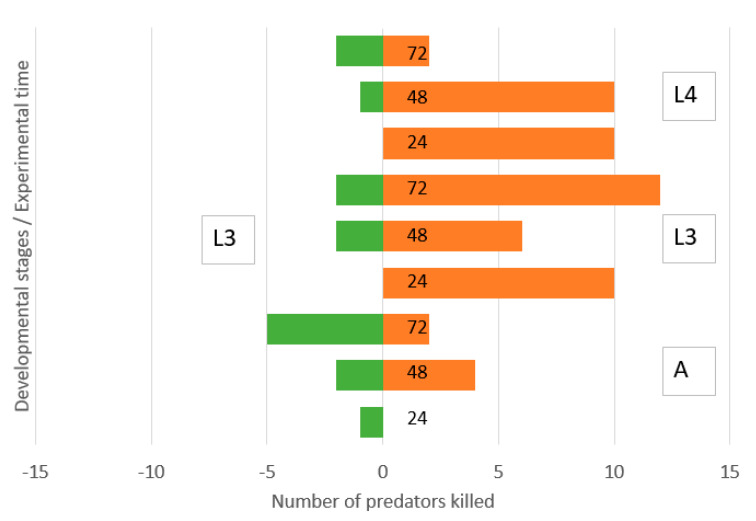
Direction of intraguild predation between *Chrysoperla carnea*, third instar larvae, and *Hippodamia variegata* adults (A), third instar larvae (L3) and fourth instar larvae (L4) in a simple arena. Numbers of predators killed of initial N = 40 during 0–24, 24–48 and 48–72 h.

**Figure 5 insects-11-00288-f005:**
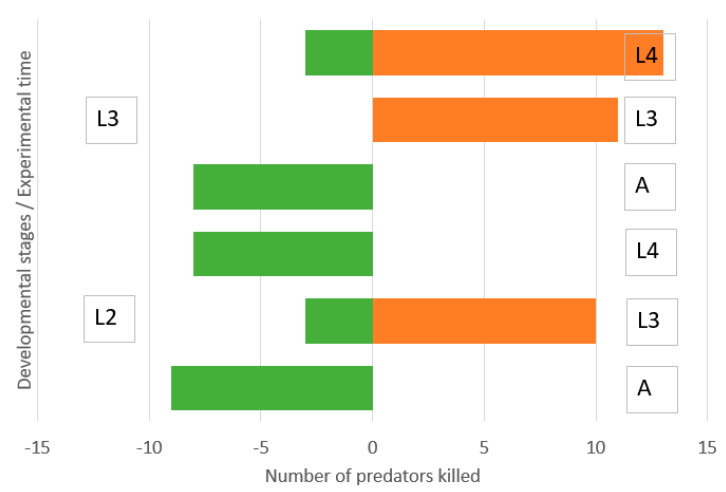
Direction of intraguild predation between *Chrysoperla carnea*, second (L2) and third instar larvae (L3), and *Hippodamia variegata* adults (A), third instar larvae (L3) and fourth instar larvae (L4) in microcosm. Numbers of predators killed of initial N = 120 during three days.

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
