# Peer review of "Intraguild Predation between Chrysoperla carnea (Neuroptera: Chrysopidae) and Hippodamia variegata (Coleoptera: Coccinellidae) at Various Extraguild Prey Densities and Arena Complexities"

_insects, 2020, doi:10.3390/insects11050288_

Round 1

Reviewer 1 Report

I find this version of the manuscript significantly improved. However, I have a series of observations and comments regarding the methodology and statistics.

lines 37-39: rephrase this sentence.

59-61: rephrase this sentence.

112-114: please explain the reason for choosing the four prey densities. The relationship between IGP and prey density is an important part of the manuscript, but I have some doubts about the results obtained by using densities that grow non-linearly. It seems obvious to me that with increasing prey, the encounter rate between predators decreases, especially if the number of preys grows exponentially.

129-131: rephrase this sentence.

142-143: it is not clear to me what has been done. Has a Tweedie generalized linear model been used? If yes, what the power variance function is?

143-145: this sentence means nothing. I ask my question again: what analysis was carried out?

145-146: Have the same data been analyzed twice? and with two different models? I hope not.

146-147: is there an a priori biological reason to use an exponential regression (or any other regression)? This point needs clarity, also in the other sections of the manuscript.

In general, the statistical analysis section needs more clarity. As a reader, I find it hard to understand how the data was analyzed. In my opinion, it would be appropriate to well specify the models used, the reasons that led you to an analysis and the factors involved in each analysis.

160-161: what is the subject of the sentence? please rephrase this sentence.

165-167: rephrase this sentence.

178-179: which variables?

185-187: it is not clear to me what kind of analysis has been carried out. Which factors?

222-224: references are missing.

228: “…and larva to larva interaction”: what does it mean?

230-231: I disagree, it does not replace the experimental control. I understand the meaning, but it must be better explained. Why should you have had a control without predators? and why isn't it important in your experiment?

Author Response

Reviewer 1

I find this version of the manuscript significantly improved. However, I have a series of observations and comments regarding the methodology and statistics.

lines 37-39: rephrase this sentence.

Reply: The sentence has been rephrased

59-61: rephrase this sentence.

Reply: The sentence has been rephrased

112-114: please explain the reason for choosing the four prey densities. The relationship between IGP and prey density is an important part of the manuscript, but I have some doubts about the results obtained by using densities that grow non-linearly. It seems obvious to me that with increasing prey, the encounter rate between predators decreases, especially if the number of preys grows exponentially.

Reply: Regarding the predation rate of C. carnea and H. variegata we tried to select three different densities of low, medium and high. Actually, we selected densities which allow the IGP occurrence and also provide sufficient food sources for the complete development of both predators. Furthermore, it has been shown that in some cases IGP reduced only at great aphid density (Lucas et al.,1998). Our set of aphid densities does not grow exponentially but roughly according to the second power. It means that the density per cm2 increases quickly if the linear density of aphids per cm connected to the walking distance increases linearly. The prey density perceived by the predator is probably closer to this linear encounter rate.

129-131: rephrase this sentence.

Reply: The sentence has been rephrased

142-143: it is not clear to me what has been done. Has a Tweedie generalized linear model been used? If yes, what the power variance function is?

Reply: It has been explained within text

143-145: this sentence means nothing. I ask my question again: what analysis was carried out?

145-146: Have the same data been analyzed twice? and with two different models? I hope not.

Reply: The three sentences have been rephrased. Moreover, when reporting to the generalized analyses in results, the word “Wald” has been replaced by chi-square.

146-147: is there an a priori biological reason to use an exponential regression (or any other regression)? This point needs clarity, also in the other sections of the manuscript.

Reply: No. The data showed relationship most resembling exponential decay.

In general, the statistical analysis section needs more clarity. As a reader, I find it hard to understand how the data was analyzed. In my opinion, it would be appropriate to well specify the models used, the reasons that led you to an analysis and the factors involved in each analysis.

Reply: All the information is present in methods and results. If anybody uses the Statistica software, he will see exactly what choices have been done.

160-161: what is the subject of the sentence? please rephrase this sentence.

Reply: The sentence has been rephrased

165-167: rephrase this sentence.

Reply: The sentence has been rephrased

178-179: which variables?

Reply: They were added to the text.

185-187: it is not clear to me what kind of analysis has been carried out.

Reply: It was repetition of previous sentence. Deleted.

Which factors?

Reply: Added.

222-224: references are missing.

Reply: Relevant references have been added.

228: “…and larva to larva interaction”: what does it mean?

Reply: Rephrased.

230-231: I disagree, it does not replace the experimental control. I understand the meaning, but it must be better explained. Why should you have had a control without predators? and why isn't it important in your experiment?

Reply: Controls without predators were analyzed in our previous study [29].

Reviewer 2 Report

The manuscript was rewritten in several parts welcoming my comments in the previous revision. Probably a reading of the text by an English native speaker would improve the style in English.

The statistical analysis has been modified. To analyze IL as function of EGP, a regression was made using the data collected in each of the 3 days. I think this is acceptable because there is no significant relationship between IL and time. My only problem is that the individual points are ratios. The size of the group on which the ratio is calculated changes with the day of the experiment since the group consists of the pairs of predators still alive. At least the average number of pairs of predators in the groups, as well as the maximum and minimum number should be reported. The number of points used for regression should also be indicated.

Author Response

Reviewer 2

The manuscript was rewritten in several parts welcoming my comments in the previous revision. Probably a reading of the text by an English native speaker would improve the style in English.

The statistical analysis has been modified. To analyze IL as function of EGP, a regression was made using the data collected in each of the 3 days. I think this is acceptable because there is no significant relationship between IL and time. My only problem is that the individual points are ratios. The size of the group on which the ratio is calculated changes with the day of the experiment since the group consists of the pairs of predators still alive. At least the average number of pairs of predators in the groups, as well as the maximum and minimum number should be reported.

This is an interesting additional information. We have added the values to Results.

The number of points used for regression should also be indicated.

These numbers are inherent to the experimental set up, but for better clarity they are added to the figure captures.

Reviewer 3 Report

The authors present the results of a fairly large and complicated laboratory study where they tested how a number of different factors influence intraguild predation interactions between two species.  The work builds off previous studies that have demonstrated the possible importance of these different effects, and it does so for a particular unique combination of species.

While this appears to be a solid research experiment, I do have some particular questions and/or concerns related to the methods and the analyses.  The answers to which could affect the paper and what it is able to say.  Second, I have some specific items that the authors should try to address to increase the strength and clarity of the overall manuscript.

Methods & Analyses

I will admit that I’m not very familiar with some of the choices for analyses, so it may be that there are places where the authors simply need to give more explanation and justification of their approach so that a broader readership can understand what they did and why.

1) Were all these experiments from the same treatments run at the same time?  Sometimes large experiments have to be done over several runs and the author includes some sort of blocking to account for variation that may occur from the different periods when the experiment is done.

2) What treatments were run at the same time?  I did not get a good sense of whether all of the different treatments were run at the same time and could, therefore, be appropriately compared directly against each other.  Alternatively, were only certain treatments (or sets of treatments) run at the same time whereas different treatments were run at different times.  When treatments or levels of treatments are completely run at different times, their comparison is traditionally confounded with any differences that occur just because of when they were run.  This issue is important in terms of knowing how things were set up, whether the statistics deals with things appropriately, and ultimately knowing how to compare various treatments.

3) The answer to this is likely related to #2, but why weren’t there more interactions tested among variables? I would have thought things like the interaction between prey density and habitat would be of interest to the authors.  Did the experimental setup and experimental analyses allow for such interaction tests?

4) It seemed like you looked at prey density and time separately (e.g., 169), but wouldn’t you expect them to interact and confound each other?  Specifically, if predators are allowed to consume prey continuously, I would expect the differences in prey density to change each day (e.g., start at 0, 50, 150, and 400, but next day it is 0, 20, 110, and 350).  I’m not sure how your analyses accounts for that. Moreover, is each day treated independently or does the analysis correctly account for the repeated measures aspect of the design?

5) Your metrics (IL, AI, and DI) are interesting and make sense, but I didn’t see where these came from.  Have these been used elsewhere or are these something the authors created.  Either way, I think a little more information could justify this approach.

Manuscript clarity

6) I am a big fan of having a parallel structure throughout the paper.  In other words, always talk about your main pieces in the same order and same way throughout.  This helps makes things much clearer.  For example, in the discussion, you have a nice structure of discussing prey density, habitat, and body size/aggressiveness.  That simple and straightforward list of topics should also be used in the introduction and in the results.  Right now, those pieces are there, but not in the same clear categories, and this it is harder to follow the ideas through different parts of the paper.  The exact structure you use is up to you, but having an explicit structure repeatedly used in the same way throughout the paper will help its clarity.

7) There are a lot of different results to follow, particularly as it relates to the three metrics.  I think addressing my item #6 will help, but I would also consider adding an additional table or visual aid to help the reader sort through everything more easily.  It might be that doing so will help add a side story about comparing and contrasting IL vs AI vs DI (not currently in the discussion).

8) The first few paragraphs of the introduction provide a lot of detail about IGP, but it isn’t always directly relevant to the paper itself (or at least it’s not clear why the reader needs to know those things).  I would consider another look at those paragraphs and try to focus on writing a paragraph or two that sets up why IGP is important and why work (like this paper) still needs to be done.  I think this will help focus the bigger picture context of the work.

9) Overall the writing was fairly good (if sometimes hard to keep all the information and ideas straight).  However, there is some additional fine scale editing that can be done once all other changes are addressed.

Author Response

Reviewer 3

The authors present the results of a fairly large and complicated laboratory study where they tested how a number of different factors influence intraguild predation interactions between two species.  The work builds off previous studies that have demonstrated the possible importance of these different effects, and it does so for a particular unique combination of species.

While this appears to be a solid research experiment, I do have some particular questions and/or concerns related to the methods and the analyses.  The answers to which could affect the paper and what it is able to say.  Second, I have some specific items that the authors should try to address to increase the strength and clarity of the overall manuscript.

Methods & Analyses

I will admit that I’m not very familiar with some of the choices for analyses, so it may be that there are places where the authors simply need to give more explanation and justification of their approach so that a broader readership can understand what they did and why.

1) Were all these experiments from the same treatments run at the same time?  Sometimes large experiments have to be done over several runs and the author includes some sort of blocking to account for variation that may occur from the different periods when the experiment is done.

 Reply: Yes, the experiment have been carried on nearly simultaneously.

2) What treatments were run at the same time?  I did not get a good sense of whether all of the different treatments were run at the same time and could, therefore, be appropriately compared directly against each other.  Alternatively, were only certain treatments (or sets of treatments) run at the same time whereas different treatments were run at different times.  When treatments or levels of treatments are completely run at different times, their comparison is traditionally confounded with any differences that occur just because of when they were run.  This issue is important in terms of knowing how things were set up, whether the statistics deals with things appropriately, and ultimately knowing how to compare various treatments.

 Reply: We tried to run the different treatments at the same time but with 3 or 4 replicates from each. Of course, it has been tried to reduce the time space between different runs in such a way that whole set of experiments being completed in one week approximately.

3) The answer to this is likely related to #2, but why weren’t there more interactions tested among variables? I would have thought things like the interaction between prey density and habitat would be of interest to the authors.  Did the experimental setup and experimental analyses allow for such interaction tests?

All this has been tested, see sentence “Then we compared the data from simple arena and microcosm …” at the end of section 3.1.

4) It seemed like you looked at prey density and time separately (e.g., 169), but wouldn’t you expect them to interact and confound each other?  Specifically, if predators are allowed to consume prey continuously, I would expect the differences in prey density to change each day (e.g., start at 0, 50, 150, and 400, but next day it is 0, 20, 110, and 350).  I’m not sure how your analyses accounts for that. Moreover, is each day treated independently or does the analysis correctly account for the repeated measures aspect of the design?

Yes, the two factors interact and the analysis would assign time as an important factor only if it had additional effect to the decreasing EGP. When all effect of time was realized through prey density, the analysis prefers to give significance to prey density. 

5) Your metrics (IL, AI, and DI) are interesting and make sense, but I didn’t see where these came from.  Have these been used elsewhere or are these something the authors created.  Either way, I think a little more information could justify this approach.

Reply: Yes, those indices have been used by other authors. Please see Lucas et al. 1998 and Lucas, 2005. One reference added.

 Manuscript clarity

6)  I am a big fan of having a parallel structure throughout the paper.  In other words, always talk about your main pieces in the same order and same way throughout.  This helps makes things much clearer.  For example, in the discussion, you have a nice structure of discussing prey density, habitat, and body size/aggressiveness.  That simple and straightforward list of topics should also be used in the introduction and in the results.  Right now, those pieces are there, but not in the same clear categories, and this it is harder to follow the ideas through different parts of the paper.  The exact structure you use is up to you, but having an explicit structure repeatedly used in the same way throughout the paper will help its clarity.

Such structure would be nice. But to mirror the logic structure of discussion in results is not possible, because the analyses we report there evaluate all the factors in a single model. Results say variable XX (e.g. IL) changed with factors A,B, not C. Thus, we had to switch the structure completely in discussion. 

7) There are a lot of different results to follow, particularly as it relates to the three metrics.  I think addressing my item #6 will help, but I would also consider adding an additional table or visual aid to help the reader sort through everything more easily.  It might be that doing so will help add a side story about comparing and contrasting IL vs AI vs DI (not currently in the discussion).

There could be even more metrics, analyses, factor interactions and so on, and we already reduced the amount of results! In previous versions of the MS, we had such tables. But they mostly included non-significant cells, occupying too much space with little positive information which was not easier to sort out. This version is the third, and we hope the most clear one.  

8) The first few paragraphs of the introduction provide a lot of detail about IGP, but it isn’t always directly relevant to the paper itself (or at least it’s not clear why the reader needs to know those things).  I would consider another look at those paragraphs and try to focus on writing a paragraph or two that sets up why IGP is important and why work (like this paper) still needs to be done.  I think this will help focus the bigger picture context of the work.

I agree, the beginning is rather a boring reading But the general overview presented there was required by reviewers of the previous version. Anyway, do scientists begin reading experimental articles from the introduction? 

9) Overall the writing was fairly good (if sometimes hard to keep all the information and ideas straight).  However, there is some additional fine scale editing that can be done once all other changes are addressed.

Yes, hopefully after acceptance of present changes we will ask a native speaker for such editing.

Reviewer 4 Report

I would say that this manuscript should be accepted for the publication of Insects after revisions because this manuscript is generally well written and it contains interesting and important results in a field of entomology and ecology.

At least following points should be cleared before the acceptance.

Major points

L104-L140 How did you distinguish between dead individuals caused by predation and those by starvation. Please describe the details in M & M. I would be strongly afraid that the experiments at 72 hours may include the data of the starvation death.

Fig. 3 & Fig. 4 The data shown here may be combined data with different aphid densities, although you admit that the prey density is important on the IGP consequences (L222-224). I would say that you should reanalyze these data and show at different aphid densities with any statistical analysis.

Minor points

L91 Please describe initial population number.

L107-108 Please indicate N(number).

L88-103 I would recommend you that you should describe the nature of C. carnea and H. variegate (preference of aphid species and natural habitat (e.g. glass, shrub, or tree), the overlap intensity of prey and habitat, breading season, et al.) for better understanding of readers.

Author Response

Reviewer 4

I would say that this manuscript should be accepted for the publication of Insects after revisions because this manuscript is generally well written and it contains interesting and important results in a field of entomology and ecology.

At least following points should be cleared before the acceptance.

Major points

L104-L140 How did you distinguish between dead individuals caused by predation and those by starvation. Please describe the details in M & M. I would be strongly afraid that the experiments at 72 hours may include the data of the starvation death.

Reply: Attacked and killed prey were punctured and their body fluids have been sucked up or they completely consumed by predators (lines 121-122). However, nobody could distinguish between an IG prey killed despite it was in good state and defended itself and a morbid prey that did not defend absolutely, except if all were recorded on a video camera all the time, which would not be possible with the high number of replications.

Fig. 3 & Fig. 4 The data shown here may be combined data with different aphid densities, although you admit that the prey density is important on the IGP consequences (L222-224). I would say that you should reanalyze these data and show at different aphid densities with any statistical analysis.

Contrastingly to IL, the IGP direction (DI) was independent on the aphid density (χ2=0.40, df=1, p=0.52). Moreover, the aphid density was a continuous, not categorical variable (it was on the beginning, but not during the experiments). We would have to split the results artificially.

Minor points

L91 Please describe initial population number.

Reply: They were collected several times but the precise number of collected predators has not been recorded.  

L107-108 Please indicate N (number).

Added, ten replications per combination.

L88-103 I would recommend you that you should describe the nature of C. carnea and H. variegate (preference of aphid species and natural habitat (e.g. glass, shrub, or tree), the overlap intensity of prey and habitat, breading season, et al.) for better understanding of readers.

Reply: Both C. carnea and H. variegata are among the most common predators of different aphid species in diverse agroecosystems including crops and orchards. Their polyphagous feeding habit and spatial co-occurrence increase their encounter rate potentially. Thus, this encourage us to investigate on possible IGP interaction between them under different situations. 

Round 2

Reviewer 3 Report

See previous review.